# Estimating orientation in natural scenes: A spiking neural network model of the insect central complex

**Rachael Stentiford**[1]*, **James C. Knight**[1], **Thomas Nowotny**[1], **Andrew Philippides**[1], **Paul Graham**[2]

**1** Department of Informatics, University of Sussex, Brighton, United Kingdom, **2** School of Life Sciences, University of Sussex, Brighton, United Kingdom

* r.l.stentiford@sussex.ac.uk

**Data Availability Statement:** All code used for this project has been made publicly available on GitHub: https://github.com/stenti/stentiford_cx_ra. Videos are available at https://doi.org/10.25377/sussex.25196528.

## Abstract

The central complex of insects contains cells, organised as a ring attractor, that encode head direction. The 'bump' of activity in the ring can be updated by idiothetic cues and external sensory information. Plasticity at the synapses between these cells and the ring neurons, that are responsible for bringing sensory information into the central complex, has been proposed to form a mapping between visual cues and the heading estimate which allows for more accurate tracking of the current heading, than if only idiothetic information were used. In *Drosophila*, ring neurons have well characterised non-linear receptive fields. In this work we produce synthetic versions of these visual receptive fields using a combination of excitatory inputs and mutual inhibition between ring neurons. We use these receptive fields to bring visual information into a spiking neural network model of the insect central complex based on the recently published Drosophila connectome. Previous modelling work has focused on how this circuit functions as a ring attractor using the same type of simple visual cues commonly used experimentally. While we initially test the model on these simple stimuli, we then go on to apply the model to complex natural scenes containing multiple conflicting cues. We show that this simple visual filtering provided by the ring neurons is sufficient to form a mapping between heading and visual features and maintain the heading estimate in the absence of angular velocity input. The network is successful at tracking heading even when presented with videos of natural scenes containing conflicting information from environmental changes and translation of the camera.

## Author summary

To navigate through the world animals require knowledge of the direction they are facing. Insects keep track of this 'head direction' with a population of 'compass' neurons. These cells can use internal measures of angular velocity to maintain the heading estimate but this becomes inaccurate over time and needs to be stabilised by environmental cues. In this work we produce a spiking neural network model replicating the connectivity between regions of the insect brain known to be involved in keeping track of heading and

**Funding:** RS, JK, AP, TN and PG were funded by EPSRC project EP/S030964/1 (https://gow.epsrc. ukri.org/NGBOViewGrant.aspx?GrantRef=EP/ S030964/1). JK was additionally funded by EPSRC project EP/V052241/1 (https://gow.epsrc.ukri.org/ NGBOViewGrant.aspx?GrantRef=EP/V052241/1). The funders had no role in study design, data collection and analysis, decision to publish, or preparation of the manuscript.

**Competing interests:** The authors have declared that no competing interests exist.

the neurons which are responsible for bringing sensory information into the circuit. We show that the model replicates the dynamics of visual learning from experiments where flies learn simple visual stimuli. Then using panoramic videos of complex natural environments, we show that the learned mapping between the current estimate of heading in the compass neurons and the features of the visual scene can maintain and enforce the correct heading estimate.

## Introduction

Maintaining a stable estimate of heading direction is essential for many behaviours across species, including complex navigation behaviours such as path integration which enables animals to return directly to the nest after taking an indirect outward path [1–3]. In insects, heading direction is tracked by 'compass neurons' (EPG cells) in the ellipsoid body (EB) of the central complex that show activity with strong directional tuning which can be tied to visual features in the environment [4] analogous to head direction cells reported in mammals [5, 6]. Unlike in mammals where head direction cells are distributed across several brain regions, dendrites of insect EPG neurons are arranged in a ring (*Drosophila*, [4]) or arc (Locust, [7]; Bee, [8]) within the EB, with adjacent neurons tuned to adjacent heading directions. There have been several previous computational models describing this circuit functioning as a ring attractor, but none of them have been tested using natural stimuli or include bioplausible filtering of the visual input. There has also been a lot of experimental work on ring neurons, in particular the visual receptive fields of two subtypes ER2 and ER4d (referred to as R2 and R4 populations in our model) which are known to deliver sensory input to the EB. This work brings together visual processing by ring neurons with *Drosophila* like receptive fields, and a spiking neural network model of the central complex ring attractor network challenged with natural visual stimuli.

Fig 1B shows the model structure comprised of 6 populations of cells, excitatory EPG and PEN cells, and inhibitory R, Δ7, R2 and R4 ring neurons. Visual input to the model arrives via the R2 and R4 ring neuron populations and PEN neurons encode angular velocity (the ideothetic estimate of heading change). The mapping between visual information and the heading representation is learnt at synapses between Ring and EPG neurons. We use the activity of the EPG cells as the model output and test the accuracy of the model by observing how well these cells estimate the ground truth heading. We test the model first on visual stimuli frequently used in experimental studies before challenging the network with natural visual scenes containing conflicting information.

Due to the shape of the EB and the topographical arrangement of head direction cells, calcium imaging of the EB can reveal a single 'bump' of activity advances around the ring structure whilst the heading of the insect changes. Additional to the EB, the insect central complex contains several neuropils including the protocerebral bridge (PB; Fig 1A). PEN neurons (specifically PEN-a neurons) in the PB also show directional turning, and conjunctively encode heading and angular velocity, producing two sustained bumps of activity within the PB, one per hemisphere [9]; [10]. These activity patterns and the known connectivity within the central complex [11–13] strongly support ring attractor dynamics similar to those in the mammalian head direction system [14–18]. Ring attractors function as winner-takes-all networks, sustaining a single bump of activity within a population. The stability of a single bump relies on interactions between an excitatory population with recurrent connections and global inhibition which prevents runaway activity.

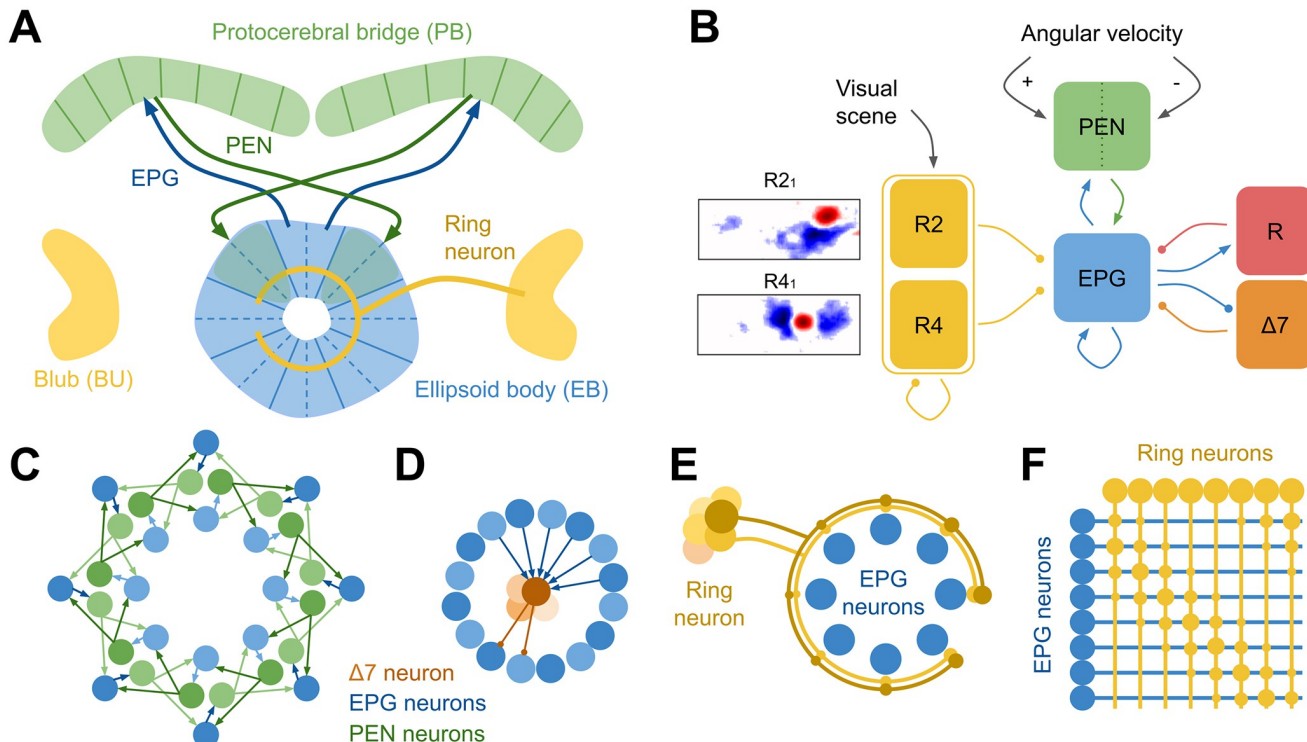

**Fig 1. Organisation of head direction circuit in the Central Complex.** (A) Major central complex regions responsible for heading tracking: ring shaped ellipsoid body (EB; blue), bulb (BU; yellow) and protocerebral bridge (PB; green). Projection patterns of two example EPG cells (blue arrows) each from one EB wedge projecting to the equivalent glomeruli (left or right) of the PB; and PEN cells (green arrows) returning projections to one tile (made up of 2 wedges) clockwise or anticlockwise further around the ring. (B) Six populations of Leaky Integrate and Fire (LIF) neurons representing cell types of the central complex. Angular velocity information is provided to PEN neurons (green), positive and negative angular velocities are delivered to the right and left PEN neurons respectively. Visual scenes are provided via R2 and R4 ring neurons (yellow). Ring neuron activation is determined by each cell's receptive field (two examples of averaged *Drosophila* ring neuron receptive fields [28] are shown, with excitatory (red) and inhibitory (blue) regions and the field of view indicated by the frame, which is 120 x 270 degrees). EPG cells (blue) send excitatory connections back onto themselves and to R (red), Δ7 (orange), and PEN cells. EPG cells receive excitatory input from PEN cells, and inhibitory connections from R, Δ7, R2 and R4 cells. (C) Effective circuit connections between each of 16 EPG (blue) and 16 PEN (green) cells split into right and left projecting cells. PEN cells return projections to both EPG cells one wedge further clockwise or anticlockwise around the ring. (D) Connections between all EPG cells and a single (out of 8) Δ7 cell. The connection pattern results in strongest inhibitory input to EPG cells opposite to the bump position. (E) Each ring neuron makes connections with every EPG cell in the ring. (F) Matrix showing all ring neuron to EPG cell connections, including strengthened weights between coactive ring and EPG cells.

Several computational models have been proposed to demonstrate how this circuit can use attractor dynamics to encode heading direction [9, 19–26]. These models all include EPG and PEN cells (or equivalent) as the excitatory component, with reciprocal connections between the two populations forming the bump by propagating activity around the ring network (Fig 1C). For the bump of activity to be maintained there must be recurrent connections back onto the currently active EPG cell. Recent *Drosophila* connectome data has revealed self-recurrent connections between excitatory EPG cells that likely ensure bump persistence [12, 13, 25]. This is in contrast to proposals in previous models where reciprocal connections between EPG cells and PEG cells would maintain the bump [19–22, 24, 26], however PEG to EPG connections are actually very weak [13]. Global inhibition is provided by two further cells types: Δ7 which functionally deliver strong inhibitory input to EPG cells on the ring that are opposite from the location of the bump (Fig 1D) [12], and GABAergic ring neurons which have characteristic ring-shaped axonal projections enabling inhibition of EPG cells around the EB (Fig 1E)

[27]. A combination of both of these inhibitory inputs has been shown to produce a more stable attractor network than either inhibitory source alone [25].

Ring neurons are also responsible for bringing in sensory information such as wind direction [29], polarized light [30], and temperature [31]. All of these sensory cues likely contribute to tethering heading estimation to the environment [32], however the heading representation can also be maintained with angular self-motion information or motor efference copies in the absence of visual information [4]. In this case the head direction estimate is subject to accumulation of error when driven purely by idiothetic cues. Therefore, understanding how sensory information takes control of the head direction signal is essential for understanding how insects maintain long-term stable estimates of heading. One proposed method of maintaining the head direction estimate is by learning relationships between visual information arriving from visual ring neurons and the EPG cells representing the current heading, such that when a visual scene is revisited the associated heading can be recalled [33, 34].

Ring neuron subtypes ER4d and ER2 have small ipsilateral visual receptive fields that have been characterised in *Drosophila* (Fig 1B) [35]. They have been implicated in a range of visual behaviours [28, 36], and as a population can encode visual features such as size, position and orientation [37]. Visual information reaches ring neurons via the anterior visual pathway [38], from the optic lobe via TuBu cells responsive to many types of visual features in the anterior optic tubercle (AOTu) to ring neurons in the bulb (BU). Similar to cells found in the mammalian visual system, each ring neuron has a receptive field formed of excitatory and inhibitory sub-fields [35].

Previous models of the insect central complex have represented visual input with several methods: simply by mapping the horizontal position of the visual cue (bright vertical bar) onto the 16 EPG cells and stimulating the appropriate PEN cell to move the bump [21, 25]; down sampling the visual scene and applying Gaussian filtering to approximate ring neuron receptive fields, then using the pixel intensity to represent inhibitory ring neuron activity [33]; and representing visual landmarks using multiple ring neuron populations each selectively tuned to the position of a specific landmark in the visual field [24, 26].

In this work we present a generalized, minimal spiking neural network model of the head direction circuit from the insect central complex including only essential connections and cell types. Input to the network comes from ring neurons with *Drosophila*-like visual receptive fields to tether the heading estimate to features of natural visual scenes. We represent visual scenes as ring neuron activity by filtering each frame with synthetic ring neuron receptive fields. The excitatory region of each ring neuron receptive field is modelled as a Gaussian function; the inhibitory regions of the receptive fields are formed through random inhibitory connections between ring neurons that decay with distance (see Methods; [13, 39]). The resulting receptive fields appear similar to those recorded in *Drosophila*.

The aim of this work is to develop a spiking model of the head direction representation in the insect central complex that can be used to explore questions about visual control over the head direction signal. We first ask if simple *Drosophila*-like visual filtering is sufficient to learn a mapping between heading and visual features, and subsequently maintain a heading estimate during rotation and simple translation. When exploring ring neuron visual input to the network, we first challenge the network with simple synthetic stimuli based on previous experimental work to test the model performance compared to known Ring and EPG neuron activity in these analogs of experiments. We then challenge the network with videos of novel natural visual scenes, and investigate how the specifics of natural environments interact with the model to determine performance.

The major contribution of this work is the investigation of how realistic visual inputs and visual processing can drive the head direction circuit. We show that using very simple yet

naturalistic visual processing based on the receptive fields recorded in *Drosophila* ring neurons, very complex visual scenes of forests and buildings can be used to build spiking representations of visual scenes associated with different headings and control the position of the compass cell activity bump. The natural scenes we present to the model are videos of outdoor spaces and included variations between frames beyond intentional translations and rotations of the camera, such as leaf movements due to wind, people walking, changes in cloud cover and movements dues to camera shake and translation.

## Materials and methods

The spiking neural network model of the insect central complex was built using PyGeNN, a Python Library for GPU-Enhanced Neural Networks [40] with a timestep of 1 ms. All neuron types were modelled as Leaky Integrate-and-Fire (LIF) neurons with membrane capacitance $C_m = 0.2nF$, resting membrane potential $V_0 = -70mV$, reset potential $V_{reset} = -70mV$, spiking threshold $V_{spike} = -45mV$, Membrane time constant $\tau_M = 20ms$, and refractory time $2ms$. Synapses between R2 or R4 neurons and EPG neurons use a custom weight update model (see below). All other synapses are non-plastic so use the standard GeNN 'StaticPulse' weight update model. All neurons shape their synaptic input using a single exponential model ('ExpCurr' in GeNN) with decay time constants of $\tau = 50ms$ for inhibitory synapses and $\tau = 100ms$ for excitatory synapses. These decay time constants were selected to produce stable circuit dynamics and are within the range of GABAb and NMDA receptors [41].

## Network structure

The connectivity included in this model is based on previous central complex models [22, 24, 25] and supported by recent *Drosophila* connectome analysis [13]. We include only essential cell types and connections in a generalised network that is not species-specific. The model includes five cell types: excitatory EPG and PEN neurons, and inhibitory Δ7, R and Ring neurons (Fig 1B).

The EB is organised into 8 tiles, with each tile subdivided into two wedges containing either a right or left-projecting EPG cell (total = 16; Fig 1A). Left and right-projecting EPG cells target PEN cells in the left and right hemisphere's protocerebral bridge (PB). The PB contains 16 or 18 glomeruli depending on species, here we include 16 PEN cells, one per glomeruli, each of which is enervated by the equivalent EPG cell (Fig 1A and 1C). PEN cells return projections back onto EPG cells in a shifted pattern (Fig 1C), targeting two EPG cells one tile around the ring either anticlockwise or clockwise for neurons in the right or left PB respectively, facilitating propagation of activity around the ring. Predictable bump shifts can be induced by stimulating PENa cells [10]. Through their reciprocal connections with EPG cells, PEN cells propagate activity around the ring network and, when left and right PEN cell firing rates are imbalanced, move the bump around the ring. Self-recurrent connections from the the currently active EPG cell ensure that the bump of activity persists. The activity of EPG cells is confined to a single bump by inhibitory inputs from Δ7 neurons [11]. Δ7 cell activity is in turn generated through excitatory inputs from EPG neurons (Fig 1D). The observed connectivity between Δ7 cells and EPG cells is overly complex than would be necessary to contain the bump and likely contributes to other mechanisms, here we use a slightly simplified connectivity pattern suited to a network with 16 EPG cells and 8 Δ7 cells that results in strongest inhibitory input to EPG cells on the opposite side to the bump (Figs 1D and 2A). Global inhibition is also provided by one R neuron which is reciprocally connected to all EPG cells (Fig 2A). See Table 1 for specific initial weights.

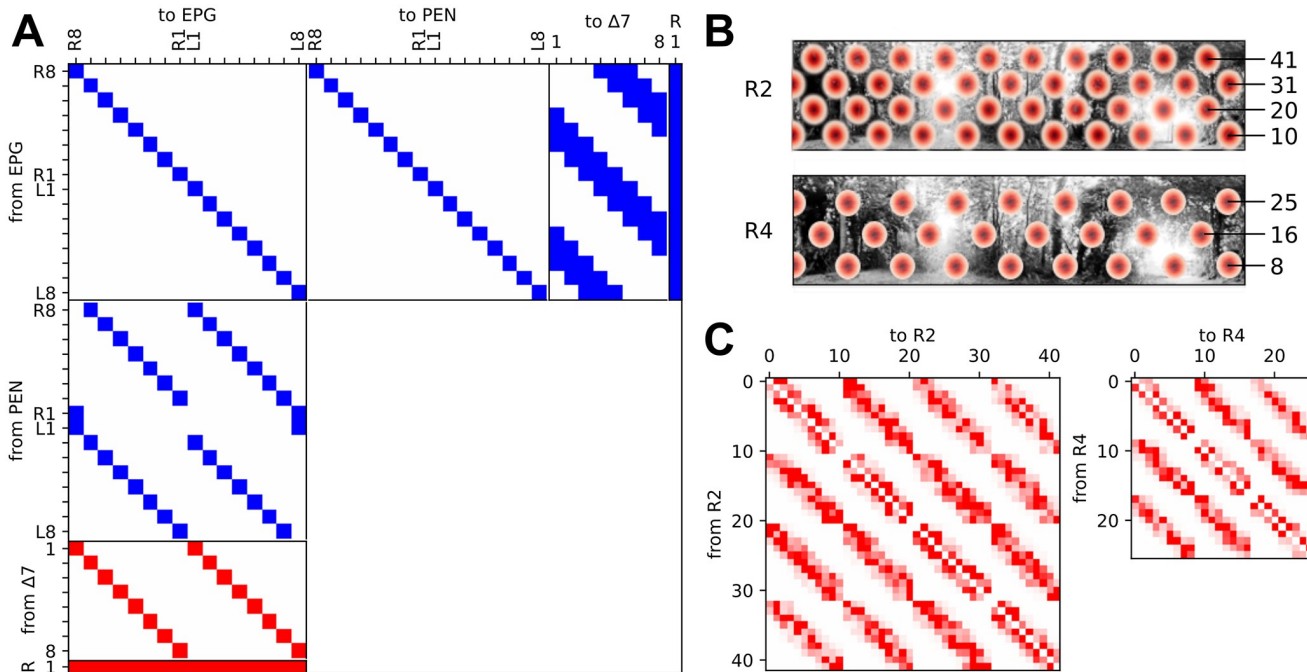

**Fig 2. Specific cell to cell connections between populations.** (A) Matrix showing connections between EPG, PEN, Δ7 and R cells, following previously determined connectivity [22, 24, 25]. Blue and red indicate excitatory and inhibitory connections respectively. EPG → EPG (weight 0.02), EPG → PEN (weight 0.13), EPG → R (weight 0.01), EPG → Δ7 (weight 0.05), PEN → EPG (weight 0.14), R → EPG (weight -1.3), and Δ7 → EPG (weight -2.6). (B) R2 (top) and R4 (bottom) ring neuron receptive field centers arranged in hexagonal pattern over the visual scene. Cells are ordered such that their receptive fields are arranged from bottom left to the top right of the scene along rows. The cell at the end of each row is labeled. (C) Inhibitory connections between ring neurons. These connections decay with distance between field centers with random scaling to produce irregular inhibitory regions of the receptive fields (see S1 Fig). R4 → R4 (max weight 0.3) and R2 → R2 (max weight 0.3). Not shown R4 → EPG (initial random weight between 0 and -0.05), R2 → EPG (initial random weight between 0 and -0.05).

## Getting the bump moving

PEN-a neurons conjunctively encode both head direction and angular velocity [9, 10]. By biasing current input representing angular velocity to one half of the PB we can disrupt the balance between input to EPG cells and drive the bump either clockwise of anticlockwise around the ring [19–22, 24]. Constant angular velocity (corresponding to rotating in one direction at a constant speed) is provided to PEN cells using a GeNN current source model [40] with constant magnitude. Current is supplied with equal positive and negative value to PEN cells in the left and right hemisphere respectively to achieve clockwise rotation and vice versa for anticlockwise

**Table 1. Connection weights between populations.**

| Connection | Weight (nA) | Connection | Weight (nA) | Connection | Weight (nA) |
|---|---|---|---|---|---|
| EPG → PEN | 0.13 | PEN → EPG | 0.14 | R2 → EPG | -0.05 |
| EPG → R | 0.01 | R → EPG | -1.3 | R4 → EPG | -0.05 |
| EPG → D7 | 0.05 | D7 → EPG | -2.6 | R2 → R2 | -0.3 |
| EPG → EPG | 0.02 | | | R4 → R4 | -0.3 |

Weights (nA) of connections shown in Fig 2 varies with cell type in order to create a balanced ring attractor. Many alternative combinations of parameters values could produce an attractor network.

rotation. This results in stronger inputs from PEN cells to EPG cells one wedge further clockwise or anticlockwise around the ellipsoid body driving the bump around the ring.

**Ring neuron input to the Ellipsoid Body.** The bump position is also manipulated by inhibitory inputs from R2 and R4 ring neurons which bring information about the visual scene. The visual receptive fields of 28 R2 and 14 R4 ring neurons characterised in *Drosophila* [35] are large and distributed unevenly across the visual field (S1A Fig), with excitatory and inhibitory sub-fields where the ring neuron activity is modulated up or down when a bright stimulus is presented (Fig 1B). Here we produced synthetic ring neuron receptive fields, 42 R2 and 26 R4 to more closely match the number of each of these cell types identified in the connectome analysis [13]. We tested three arrangements to tile the 95 × 360 degree visual scene: hexagonal grid, square grid and random placement (for random arrangement, half the RFs were distributed randomly in one half of the visual field and reflected onto the other half). We found that the broad pattern of results was independent of the arrangement of receptive fields (Fig 3G and S1 Fig) and so for all results shown, unless otherwise stated, a hexagonal arrangement of receptive fields was used (Fig 2B). Each ring neuron has an excitatory RF only, with the inhibitory regions forming through mutual inhibition between ring neurons of the same type [13](Fig 2C).

The strength of the excitatory receptive field is modelled as a Gaussian function ($\sigma = 225$) scaled to deliver a current input of between 0 and 0.35 $nA$ to the ring neurons. Ring neuron activity is generated by applying the synthetic receptive fields as filters to 95 × 360 degree panoramic video frames and using the resulting activation value as current input to ring neurons via a GeNN current source model. The weight of the inhibitory connections between Ring neurons depends on the distance between the RF centers (Gaussian function $\sigma = 40$), and a random weighting (Fig 2C) which ensures nonuniform inhibitory input and irregular center surround receptive fields as observed in *Drosophila* (S1 Fig).

This method of generating the inhibitory regions through mutual inhibition is inspired by the presence of inhibitory connections between ring neurons highlighted by connectome analysis [13]. However similar receptive field organisation has been observed in upstream TuBu cells [42]. Therefore some aspects of the fly ring neuron receptive field properties are likely inherited from the upstream cells. However the characteristic receptive field shapes previously observed in ring neurons [35] do not appear fully formed in the TuBu receptive fields, and may still be refined by inhibitory connections between ring neurons. The inhibitory connections are mostly specific within ring neuron type [39], and inhibitory regions *Drosophila* ring neuron receptive fields appear similar to the sum across neighbouring excitatory regions of other cells with some random weightings. Using this method to generate synthetic receptive fields allows us to use the full number of visual ring neuron identified by connectome analysis, and to explore how the arrangement of visual inputs to the EB may contribute to targeted learning through selective attention.

## Synaptic plasticity

Each ring neuron synapses with every EPG cell in the ring (Fig 1F; [27]). To map visual cues represented by the activity of R2 and R4 ring neurons onto headings represented by EPG neurons, Anti-Hebbian synaptic plasticity is enabled at these synapses [33, 34] (Fig 1E and 1F). As ring neurons are inhibitory, coincident activity at these synapses triggers a change in synaptic weight ($w_{i,j}$) towards $w_{max} = 0$, weakening the inhibitory synapse. The order of this activity, be it pre- then postsynaptic or post- then presynaptic does not matter. Non- coincident spikes trigger a weight change towards $w_{min} = -0.3$ increasing the strength of the inhibitory synapse. We implement this using a Spike Timing Dependent Plasticity (STDP) rule inspired by the

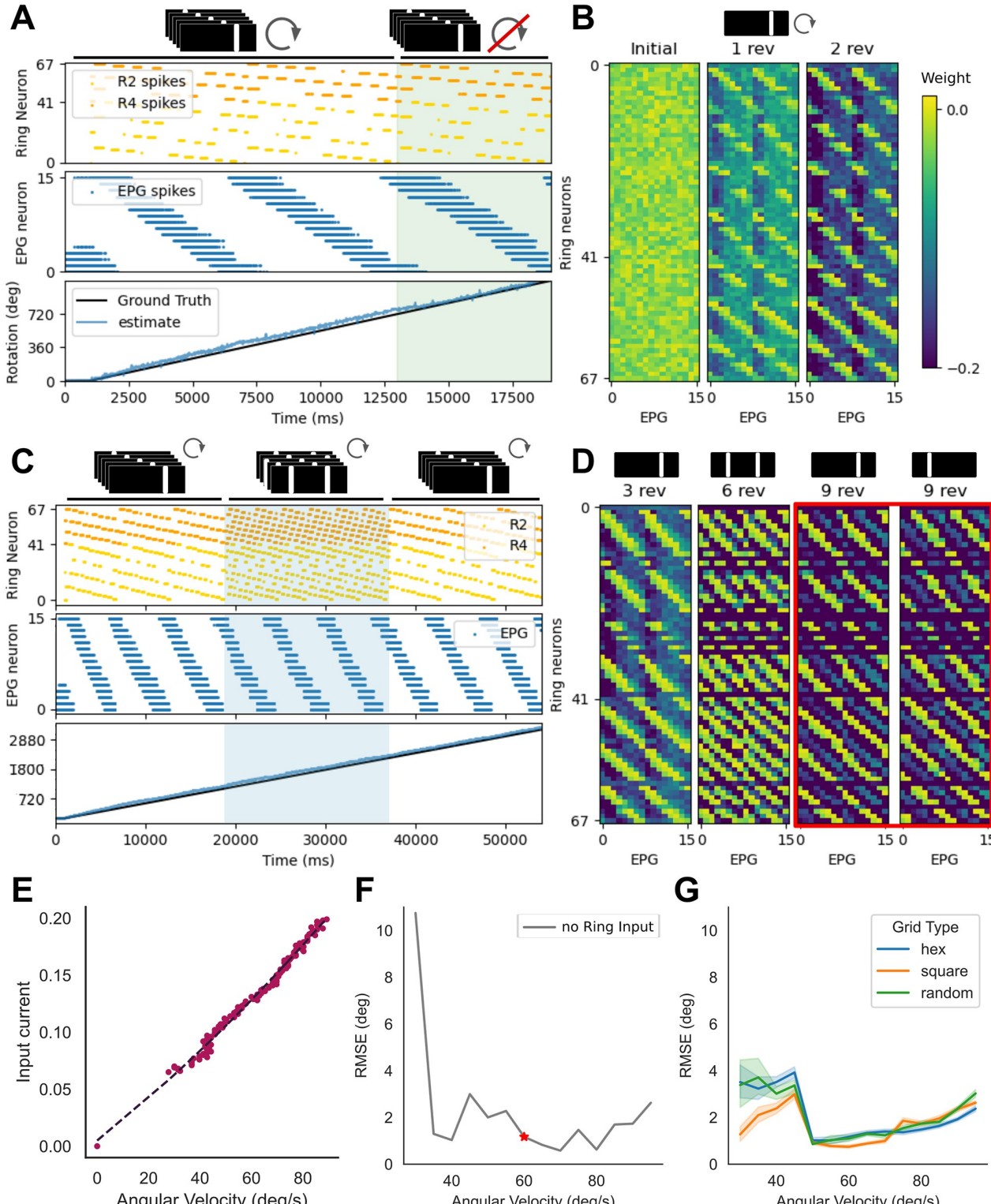

**Fig 3. Bump dynamics for vertical bar visual scenes.** (A) Raster plots of ring neuron and EPG cell spikes over 3 revolutions with a visual scene of a single bright vertical bar. For two revolutions both visual and angular velocity input are provided to the model. Only visual input is provided during the final revolution (green shaded region). The unwrapped ground truth heading and estimated heading (EPG cell at the center of the bump) are also shown. (B) Evolution of the synaptic weight matrix from random initial weights after 1 and 2 revolutions for the one bar visual scene. (C) Raster plots of ring neuron and EPG cell spikes over 9 revolutions. The initial 3 revolutions with a visual scene with a single bright vertical bar. For a further 3

revolutions an ambiguous visual scene with two identical bright vertical bars separated by 180 degrees (blue shaded region). For the final three revolutions the single bar visual scene was returned with the bar at one of the two offsets. A single bump of activity is maintained throughout by ring attractor dynamics. (D) Evolution of the synaptic weight matrix from random initial weights after each visual scene presentation. When two bars are presented, both positions are represented in the weight matrix. The final weight matrix can toggle between two states depending on the offset position of the final vertical bar (red box). Either maintaining the original mapping (3 revolutions vs 9 revolutions left) or remapping to the new bar position (3 revolutions vs 9 revolutions right). (E) Input current to PEN cells required to produce bump movement of different angular velocities. Below input current of 0.06 bump movement cannot be produced. (F) RMSE over the full 3 rotations when only input to PEN cells is provided for a range of angular velocities. Red star shows baseline RMSE for the angular velocity used in the 3 revolutions experiments. (G) RMSE during probe trial of the 3 revolutions experiment at a range of angular velocities. Hexagonal, Square Grid and Random symmetrical receptive field patterns all result in similar performance.

symmetric shape presented in [43] and implemented as a GeNN custom weight update model. The following weight updates are triggered by ring or EPG neuron spikes:

$$\Delta w_{i,j} = \eta \left( \exp\left( \frac{-|t_{pre} - t_{\text{post}}|}{\tau} \right) - \rho \right)$$

where $t_{post}$ and $t_{pre}$ are the time of the postsynaptic and presynaptic spikes respectively, $\tau = 50.0$, $\rho = 0.06$, learning rate $\eta = 0.01$. A parameter search over $\eta$ and $\rho$ (S3A Fig) showed a region of parameter space where combinations of $\eta$ and $\rho$ result in low error across all natural scenes (see below). Values were selected from this low error region. For these learning parameters and angular velocity of 60 deg/s, the weight space develops quickly, becoming stable after approximately 5 revolutions (S3C and S3D Fig).

## Visual stimuli

We first replicated visual stimuli previously used in *Drosophila* experiments [33, 34, 44], including a white vertical bar (95 × 15 degrees) on a black background, or two white vertical bars (each 95 × 15 degrees) with 180 degree separation.

We then presented the same network with a total of 33 panoramic videos of natural scenes (S2 Fig) recorded at various locations across the University of Sussex campus and adjacent Stanmer Park (S2 Fig). For 23 of these scenes, panoramas were captured as the camera was rotated in place at 10 rotations per minute, and frames were provided as visual input to ring neurons. 17 videos were recorded in densely wooded areas and 6 in open areas in different weather conditions. Videos were captured using a Kodak PIXPRO 360 4k camera and all frames were cropped and resized to 95 × 360 px and converted to greyscale. Global histogram equalization was applied for contrast adjustment using the OpenCV library. Rotating datasets were produced by mounting the PIXPRO camera onto a stepper motor programmed to rotate clockwise for 3 revolutions at 10 rotations per minute using an Arduino.

The further 10 scenes were captured using a cable driven camera similar to a spidercam or skycam (S2 Fig) and processed in the same way. The camera was moved through a 200mm radius circle, and the resulting panoramas artificially rotated to maintain heading in line with the direction of movement. The cable-driven camera consisted of a 1 × 1 m frame with 4 nema17 47mm stepper motors arranged at each corner, each with a spool of braided fishing line. The fishing line is threaded through eyelets 450mm above the frame attached to transparent acrylic arms then connected at a central point beneath the PIXPRO camera which is counterweighted. By spooling in or out each cable the camera can be moved in 3 dimensions anywhere within the frame. The ground truth position of the camera in a lab environment was recorded using a Vicon motion capture system, that tracks the relative positions of IR reflective markers attached to the camera. Assuming heading is in the direction of movement, the appropriate angle of each frame was estimated from the smoothed ground truth trajectory, and used

to artificially rotate panoramic videos after collection. Videos were downsampled to match angular velocity in both the rotation and circling scenes.

## Systematic parameter search

To investigate the robustness of our central complex model to varying parameters we systematically explored the parameter space, focusing on the synaptic weights between populations of cells. Parameter sets were considered viable if they met the following conditions: the bump must persist for the full simulation with spikes occurring in both the first and last 100 ms; the bump must be at least 2 cells wide at its narrowest and no wider than 8 cells at its widest; and the bump must move at least one full rotation within a 20s simulation. For most parameters a wide range of values could produce viable bump movement. However the weight between EPG cells and the inhibitory R cell has a large impact on bump movement. When set at the optimal value found in the initial grid search (0.01), multiple parameters sets give viable results. S4A Fig shows the number of rotations produced by each viable parameter set when varying R to EPG, Δ7 to EPG, PEN to EPG and EPG to PEN weights. For these results, EPG to R, EPG to Δ7 and EPG to EPG weights are set to constant values specified in Table 1. Viable parameter spaces were found for values between 0.01 and 0.2 for EPG to Δ7, and 0.001 and 0.8 for EPG to EPG (the widest range of any parameter). However, aside from this one value of 0.01, only one other EPG to inhibitory R cell weight (0.015) can produce consistent bump movement and only then for a small range of parameters (S4B Fig).

## Results

Here we show the behaviour of our spiking neural network model of the insect central complex with inputs from ring neurons with *Drosophila* inspired receptive fields. Using a computational approach allows us to interrogate the network using more complex natural visual input than would be possible during electrophysiological or calcium imaging experiments. To demonstrate the robustness of the network we begin with simple visual stimuli (bright vertical bars) that have been used extensively in experimental work. As we are interested in natural scenes containing conflicting information, we then presented the network with an ambiguous stimulus to observe the networks response to conflicting cues (two identical bright vertical bars [33, 34]). Finally we present natural scenes to the network, first rotating the camera on the spot (mimicking how the synthetic visual cues are presented to flies) and then translating through a circle. Most of these experiments follow the same general structure. As we assume this network encodes the mapping between heading and visual features quickly, both the visual input to ring neurons and equivalent angular velocity input to PEN cells were provided for two rotations, and then visual input only for a third rotation (probe trial).

Throughout the results, we demonstrate the ability of the network to robustly maintain a heading estimate in response to a variety of visual stimuli including bright vertical bars and natural panoramic scenes, including stimuli width competing cues. We assess performance based on how closely the center of the bump of EPG activity follows ground truth head angle (see Methods for details). Interestingly, across all experiments the bump of EPG cell activity spans approximately 4–5 cells at any given time, with each of the 16 EPG cells representing approximately 22.5° and a total bump width of between 90°- 112.5°–equivalent to 25–31.25% of the ring. This is very similar to the spread of activity observed in Calcium imaging of *Drosophila* EPG cells [4].

To measure how closely the model tracked heading using visual features, we compared the estimated heading to the ground truth heading. The estimated heading was found by binning spikes into 16.7 ms bins (time window of one frame), finding the cells active during each bin

and selecting the EPG cell at the center of the active cells, multiplied by the angle represented by each cell (22.5 degrees). The root mean squared error between the estimated heading and ground truth for the final rotation with only visual input available was used to measure model performance over a total of 400 simulations (25 random initial weight spaces between ring neurons and EPG neurons and 16 shifts in initial panorama angle).

## Learning the mapping between simple stimuli and heading

We first replicated two experimental setups used to challenge insects. To observe the learning behaviour of the network when presented with a simple stimulus we used the experimental set-up of [33, 34] and presented the network with a 15 degree wide bright vertical bar on a black background. The bar moved across the visual scene at constant angular velocity for 3 full rotations (Fig 3A). The bar stimulus passed through all ring neuron receptive fields, and the bump of activity was moved by angular velocity input to the PEN cells, resulting in co-activation between a subset of ring neurons and EPG cells at each heading. Learning at ring to EPG cell synapses results in depression between co-active cells. Reactivation of these ring neurons results in reduced inhibition to the co-active EPG cells. Angular velocity input was provided to the network for the first two rotations only (Fig 3A, white shaded region). During the third rotation (probe trial), the angular velocity input to PEN cells was turned off and the bump position is driven purely by dis-inhibition from the visual ring neurons (Fig 3A, green shaded region; RMSE 1.17 degrees). Learning at synapses between the ring neurons and EPG cells during the first and second rotation resulted in a weight matrix with clearly visible associations between active ring neurons representing the bar position and heading (Fig 3B).

We ran this experiment for a range of angular velocities, starting at 30 deg/s which is the slowest bump movement that could be reliably generated using this parameter set (Fig 3E). We measured the angular velocity produced by different current inputs to the PEN cells, and, by fitting a curve to this data, estimated the current required to move the bump at different angular velocities. As this mapping is not perfect, when only input to the PEN is available RMSE of bump position across the full 3 rotations is variable but low (Fig 3F), with error accumulating over the three rotations. With the addition of visual input via ring neurons, RMSE during the probe trial is similar for all three receptive field arrangements across angular velocities (Fig 3G). At high angular velocities, less learning is able to take place before the probe trial, resulting in diminished performance. At low angular velocities learning occurs rapidly but bump movement is slow and there is more opportunity for developing unhelpful or overly strong mappings between ring neurons and EPG cells. This highlights the need for a variable learning rate at different angular velocities, evidence for which has been observed in *Drosophila* [45].

As we are interested in natural scenes with conflicting information, following the method of [19, 34] we introduced a second conflicting cue (two bright bars separated by 180 degrees; Fig 3C) and observed changes in the weight matrix. When the network is trained with these ambiguous stimuli, a second representation of the bright bar is learned in the ring neuron to EPG weight matrix (Fig 3D). Ring attractor dynamics ensure only a single bump of activity is maintained in the ring. When returning to a single bar stimulus only one of the two competing representations in the ring neuron to EPG weight matrix is maintained, depending on which bar remains (Fig 3D, red box). A similar phenomenon has been recorded experimentally in *Drosophila* [33] and these results indicate that the network can successfully form an appropriate, flexible mapping between visual cues represented in the activity of ring neurons and head direction cells. Contrary to previous work, our model maintains an accurate representation of the ground truth when presented with ambiguous stimuli rather than skipping over half the

population when the cue is revisited [33]. While, in natural scenes with conflicting information, it is important to maintain a consistent representation, this behaviour indicates that visual cues may be less dominant in our model than observed experimentally in *Drosophila*.

## Learning a mapping between natural scenes and heading

Now that we have shown that the network is able to learn a mapping between ring and EPG neurons for simple stimuli, we applied the network to more challenging natural scenes. We presented two types of natural scenes: videos mimicking an insect rotating on the spot as in the virtual bar experiments; videos where the camera was moved in a circular trajectory.

To assess the specificity of the ring neuron representation of the visual scene we calculated correlations between either pixel values or ring neuron activations for each pair of frames, by first normalising the frames (using OpenCV's normalise function), then finding the Pearson correlation coefficient. Correlations between pixel values (Fig 4B and 4G) or ring neuron activations at each frame (Fig 4C and 4H) are highest after exactly 1 or 2 full rotations (separated by 360 and 720 degrees rotation). In most scenes correlations were low between frames at intermediate headings but with occasional raised values due to ambiguity and aliasing in the natural scenes. For instance for the scene in Fig 4A this occurs at around 180 degrees offset (faint light blue diagonal lines; Fig 4B and 4C) corresponding to gaps in the tree cover at either end of a path. Similar non-zero values can be seen for the scene in Fig 4F although it is less clear what features contributed to the visual aliasing in this scene (See S2 Fig for example frames from all scenes). However these artefacts do not lead to similar enough ring neuron activations to confound the model performance.

In both of the examples shown in Fig 4—one rotating in place and the second moving though a circular trajectory—the network successfully tracked heading during the probe trial. Ring neuron activations for the first example scene uniquely represented each heading (Fig 4C; rotation scene 5). Over the first two rotations a mapping between co-active ring and EPG cells was learned (Fig 4E) such that ring neuron input was sufficient to appropriately move the bump and maintain an estimated heading close to ground truth during the probe trial (Fig 4D; mean RMSE over 20 simulated seeds and 16 panorama shifts = 0.854 ± 0.014 degrees). For some visual scenes, including the second example (circling scene 6, Fig 4F), the mapping between ring neurons and heading contained conflicting representations and no obvious directional cue. This resulted in more error in bump movement (Fig 4J and 4K; mean RMSE over 20 simulated seeds and 16 panorama shifts = 2.694 ± 0.214 degrees). Across repetitions of the simulation higher variance in RMSE was observed.

Surprisingly, there was little difference between the rotation only and circling datasets. We expected that introducing some proximal objects and resulting parallax by placing the camera near the ground or close to trees and undergrowth where there were lots of obstructions would make these scenes especially difficult to track, but heading was tracked robustly.

As the model is capable of tracking heading for both the rotation and circling data sets, we were interested in the robustness of our model and in particular, whether forming a good mapping is more difficult for some scenes. What features of natural scenes might be the most salient or make these scenes easily discriminable? Is a behavioural strategy required to improve performance? Is it beneficial to have many ring neurons representing a scene? Do multi-directional ring neurons reduce network performance? To probe these questions we ranked the performance of the model for each natural scene based on the mean RMSE between ground truth and estimated heading for the probe trial over 20 simulated seeds each for 16 starting orientations within the panorama (320 total simulation runs), and divided natural scenes into three classes: those with high variance in the error measured across all simulations (standard

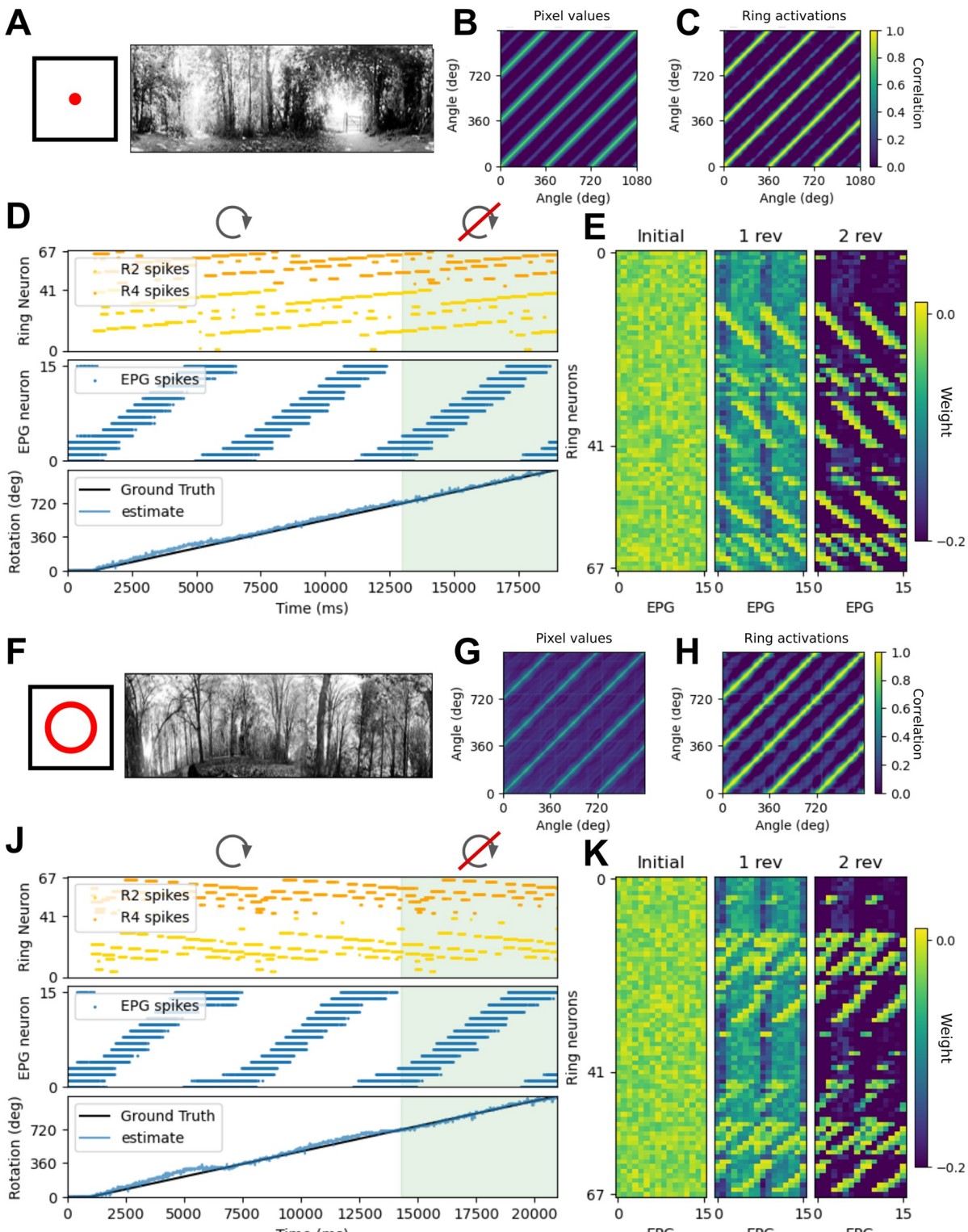

**Fig 4. Bump dynamics for revolutions in two natural visual scenes during rotation and translation.** (A,F) Single frame from each of the greyscale panoramic videos. Histogram equalisation is applied to each downsampled video (see Methods). A: Frame video with camera rotation only. F: Frame video with circular camera translation and rotation. (B,G) Correlation matrices showing correlation between each frame for three rotations. Strong correlation is seen between frames at the same rotation angle, but this correlation is not due to natural noise in the video. (C,H) Correlation matrices showing correlation between ring neuron activations at each frame for three rotations. Strong

correlation is seen between activations for frames at the same rotation angle. (D,J) Raster plots of ring neuron and EPG cell spikes over 3 revolutions, and the unwrapped ground truth heading and estimated heading (EPG cell at the center of the bump). For two revolutions both visual and angular velocity input are provided to the model. Only visual input provided during the final revolution/probe trial (green shaded area). (E,K) Evolution of the synaptic weight matrix from random initial weights after 1 and 2 revolutions.

error > 0.1), those with low variance (standard error < 0.1), and scenes with greater than 1% (3/320) failed simulations (Fig 5A). Simulations were labeled "failed" if no spikes were recorded in the last 100ms indicating the bump of activity collapsed and no heading estimate could be measured. Example frames for each video, labelled with their performance group, can be found in S2 Fig. Inspection of the panoramas in the high failure group did not reveal any obvious environmental patterns such as proximity to landmarks, sun visibility, camera shake, or density of forest, which could explain why these scenes performed worse than others.

As there were no obvious environmental features to explain the differences between high variance, low variance and failure groups, we compared these groups across several measures of ring neuron activity and image properties. Scenes in the fail group had on average higher ring neuron mean firing rate than both the low and high variance groups (ANOVA with Tukey's HSD; Low p = 0.018; High p = 0.037). High firing rate in scenes prone to failure suggests an active strategy for adapting peak ring neuron activity may be required to make these scenes more easily discriminable. High variance scenes had fewer total ring neurons active than the low variance and fail group (ANOVA with Tukey's HSD; Low p = 0.007; Fail p = 0.002). Fewer activated ring neurons indicates these scenes had less prominent or small features, resulting in fewer strong mappings between ring neurons and EPG cells, and overall less power to disinhibit EPG cells and drive bump movement.

High variance scenes also showed higher mean correlations across both correlation matrices than the low variance group (ANOVA with Tukey's HSD; mean activations correlation p = 0.019; mean frame correlation p = 0.031). This indicates higher correlations at intermediate headings, suggesting there was more directional ambiguity in the visual scene with ring neurons representing multiple parts of the visual scene. In all groups there were examples of natural scenes where individual ring neurons were active for multiple headings resulting in multi-directional tuning, another indicator of directional ambiguity. However, the presence of multi-directional cells in high variance, low variance, and failure groups indicates similar patterns of co-active ring neurons at multiple headings, rather than individual ring neurons active at multiple headings. It is the latter that results in more variable performance, because scenes are less discriminable both at the pixel level and after the large scale filtering by ring neurons.

## Discussion

We have presented a novel spiking neural network model of the insect central complex, incorporating known connectivity [12, 13] between ring neurons [35] and the ellipsoid body compass neurons which allows for the formation of a flexible mapping of visual cues onto headings. The model was able to replicate fly performance and network dynamics across a series of experimental assays with simple visual information in the form of bright vertical bars [33, 34]. When faced with ambiguous stimuli such as two vertical bars, the network forms multiple representations of the cue and stabilises on one of these when one of the two cues is removed, thus replicating the weight space dynamics proposed by [33].

Having demonstrated its plausibility, our goal was to go beyond a proof of concept and challenge the model with complex real-world visual input to see if it could sustain a stable activity bump in the ring attractor. To this end, we used panoramic videos that include natural

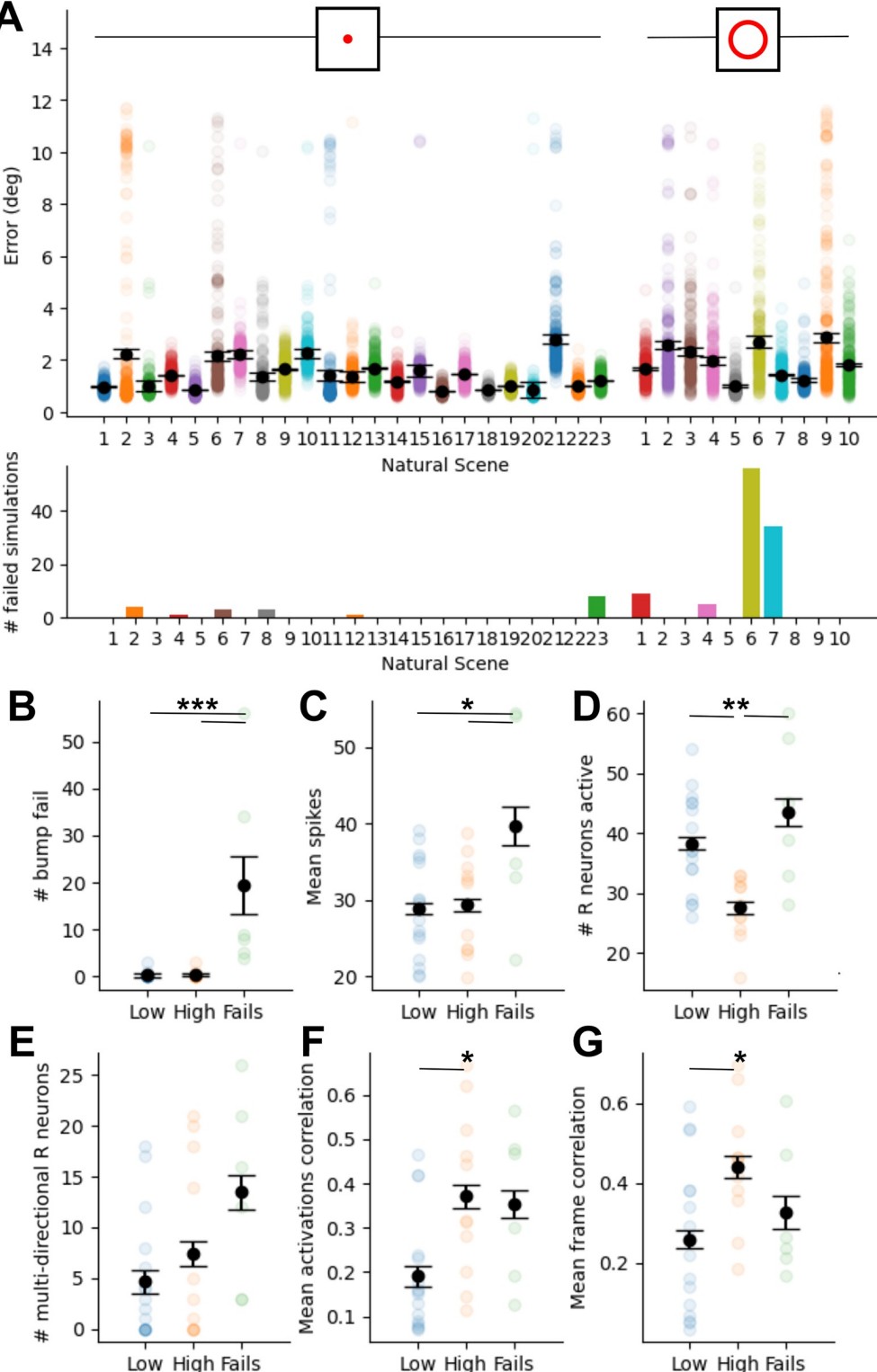

**Fig 5. Performance across different natural scenes during rotation and translation.** (A) Mean RMSE (degrees) between ground truth and estimated heading during the probe trial with only visual information available, over 20 simulated seeds and 16 shifts of the panorama initial angle ± bootstrapped standard error (top), and total number of failed simulations for each natural scene, where the bump of activity is lost (bottom). Scenes are separated into camera rotation only (left; 23 scenes), and circular translation and rotation (right; 10 scenes). (B-G) Comparisons of spiking

and image properties between natural scenes with more than 1% failure rate, high or low variance. (B) Number of failures. (C) Mean ring neuron spikes. (D) Total number of ring neurons active. (E) Number of ring neurons with multi-directional receptive fields. (F) Mean over ring neuron activation correlation matrix. (G) Mean over video frame correlation matrix. (For all plots *** ANOVA p<0.001; ** ANOVA p<0.01; * ANOVA p<0.05).

environmental variations (due to wind and lighting changes) and camera movement. We found that the information encoded by the simple visual filtering of a small number of ring neurons with *Drosophila*-like receptive fields is sufficient to robustly maintain an estimate of heading despite the complex visual input. In the following sections we discuss the implications of these results.

## Capturing the ring neuron to central complex circuit

As our focus was on how visual ring neuron inputs influence the network rather than specifically how the bump is formed or maintained, we constrained the model to the essential underlying architecture and connections required to produce the expected ring attractor dynamics, whilst using cell types and connections from up-to-date connectome data [12, 13]. Notably, PEG cells were excluded which were instrumental in previous work for maintaining bump persistence [19–22, 24, 26], because their function was replaced by direct self-recurrent connections between EPG cells identified in the connectome analysis [13]. As the bump properties remain similar in our model, we now need to look to experiments as to whether PEG cells and self-recurrent EPG activity are redundant mechanisms or whether each has its own influence over the circuit.

Secondly, to explore the transformation from visual input to neural code via the ring neurons, we interrogated the connectome to see if there were connections that could create the receptive fields previously described by [35]. We model each ring neuron with only an excitatory receptive field and propose that the function of the previous unexplained mutual inhibitory connections between the ring neurons [13, 39] is to form the functional inhibitory regions recorded in *Drosophila* [35]. Mutual inhibition prevents neurons with overlapping receptive fields from representing the same cue, helps selecting more prominent visual cues and suppresses noise arising from less activated ring neurons. Although here we only include inhibition within ring neuron type, some connectivity between ring neuron types has also been identified, which could enforce a suppression hierarchy when more useful sensory information (such as polarized light or sun position as represented by other classes of ring neuron) win out over less strong cues [13]. These ideas provide interesting testable predictions for future work.

In simplifying the circuit we made several assumptions that limit the biological completeness of the model, but importantly the simplifications are not crucial to the questions explored. For instance, cell types with no clear function and ring neuron subtypes carrying information about other sensory modalities were excluded from the model. We also used a single neuron to provide global inhibition to the network, whilst in reality, this is likely to be provided by a population of ring neurons which may be the same cells as those bringing sensory information into the network [25]. This dual role of ring neurons aligns with the experimental data where the activity of ring neurons is modulated up or down with respect to stimuli passing through the excitatory and inhibitory regions of the receptive field [35]. This raises questions about how this type of activity would impact the mapping between visual cues and the heading estimate. To incorporate this type of ring neuron coding, an alternative learning rule may be required, as we used a simple learning rule that does not take into account the order of spikes or their precise timing.

We made further assumptions regarding the arrangement of ring neuron visual receptive fields. Hexagonally arranged ring neuron receptive fields allow the network to use all available cues across the visual field, and are appropriate for keeping track of large, low-frequency visual cues [36]. Robust model performance shows the arrangement of ER2 and ER4d ring neuron receptive fields as they are observed in *Drosophila* (clustered along the horizon to the left and right of the visual field; [35]) is not essential for tracking heading in this particular task. However, this arrangement may be useful for tracking natural objects or specific environmental features. It may also be the optimal position for extracting visual information relevant to the control of avoidance or attraction behaviours [47]. Other ring neuron subtypes sensitive to polarized light have receptive fields in the upper third of the visual field [48]. Recent work also calls into question the shape of ring neuron receptive fields reported by Seelig and Jayaraman [35]. Based on the dendritic arrangement of upstream MeTu cells in the medulla of the optic lobe [48], the receptive fields of ER4d neurons may be long vertical stripes across the visual field rather than the classic centre-surround type. However centre-surround receptive fields could be achieved for different upstream morphology with a more complex inhibitory connectivity between ring neurons.

Indeed, inhibition is important in the model. The inhibitory connections between ring neurons leads the model to track visual heading with a simple form of selective attention. Varying the strength of mutual inhibition between ring neurons, or additional visual processing upstream in the anterior visual pathway, could also provide more structured visual input to the network to make salient features more prominent or task specific.

## Towards an understanding of the circuit in real-world insect behaviour

In the real world, insects need to know when to learn, but also when not to learn. For our model, the average error across all scenes remained small (below 2 degrees) regardless of whether learning remains on or off during the probe trial, although performance improved when learning was turned off (S3B Fig). However, in insects learning rates are not static. For instance, learning at the ER→EPG synapses is modulated by dopamine signalling which actively increases synaptic plasticity when angular velocity is large. This prevents over-learning of head directions at slow speeds that otherwise would be over-sampled [45] and would be a valuable future addition to the model.

It is also interesting to question what features of the visual scene insects attend to. With its simple implementation of biomimetic visual processing focused on luminance-defined cues, the network currently tracks areas of the scene with the highest intensity of light such as the sun position or visible areas of sky between trees. Ring neurons have been previously shown to respond to bright sun-like objects or bright areas of a sky gradient which are sufficient for stabilizing the head direction signal in flies [49]. However, it is important to note that although our model can use the sun to track heading on a sunny day, it is not required, and the network was also able to track heading for the same natural scenes on overcast days (S2 Fig). Furthermore, the noise inherent in the videos of natural scenes was almost entirely removed after filtering the input by ring neuron receptive fields. However, while attention may be driven by one overriding cue—light level in this instance—heading tracking still requires a population of ring neurons. By comparing natural scenes that are easy or difficult for the network to track, it is clear that rather than headings being represented by the single most active ring neuron, headings are encoded by a combination of ring neurons, each of which may be included in other combinations for other headings. This is evidenced by the appearance of multi-directional cells in all performance groups where individual ring neurons are active for multiple headings.

Natural scenes contain more information than just light intensity. Motion-defined cues are also represented by ring neurons, specifically ER4d cells which are sensitive to both motion and brightness [50]. Extending the model to include more of the visual pathway would allow us to build more complex representations of visual scenes and observe how this impacts the network performance. However, it is important to remember that visual information is not the only sensory information available to the fly to keep track of heading, and there are further ring neuron types, which bring in information about other sensory modalities to the central complex [29–31, 38]. This raises the question, how these different sensory modalities are integrated to produce the most accurate estimate of heading. A total of 46 EPG cells were identified in the *Drosophila* connectome [13], of which we currently include 16. This is almost enough cells to have 3 complete rings. This raises the intriguing possibility of having a multi-ring model in which multiple ring attractors act as redundancy in the network or alternatively, each ring may receive different sensory information, brought in by subtypes of ring neurons, and integrating over these inputs would produce a multi-sensory heading estimation.

### Implications for robotic applications and future experiments

Having shown that the model can function robustly with naturalistic input, this raises interesting implications for autonomous robotics systems that would need to maintain estimates of pose or heading in complex natural environments [51, 52]. Studying biological systems in this way highlights the potential for low-energy dedicated biomimetic circuitry, which could be plausibly implemented on neuromorphic circuits for low-power autonomous robotics applications [53–55]. Insect-inspired circuits are a particularly promising avenue for this approach as, while they solve the same problems as mammals (like tracking heading with compass neurons), they do so with far fewer neurons. Robots can also be valuable test-beds for biological hypotheses [56] and we have previously used them to elucidate models of mammalian navigation [57].

One of the most interesting implications of our model is its potential to scale. For example, one could investigate duplicating the ring of EPG neurons for multi-sensory integration, with each ring dedicated to a different behavioural task or sensory modality. Furthermore, different insects are specialists for different behaviours, and although we know the broad architecture of the central complex is conserved across species [58] there must be differences in the circuit that are dedicated to specialist behaviours. Because the majority of experimental data—particularly optogenetics and connectome studies—have focused on *Drosophila* [10, 11, 13, 30, 35, 38, 46, 48], the model presented in this work and most other previous models are biased towards the *Drosophila* circuit [9, 21, 24, 25]. However the real power of computational approaches is that they allow us to explore questions about other insect species and investigate how behavioural specialisms may come about from this conserved circuitry.

### Supporting information

**S1 Fig. Ring neuron receptive fields.** (A) 14 R2 and 7 R4 averaged Drosophila receptive fields [28]. (B) 26 synthetic R4 receptive fields with hexagonal spacing. (C) 26 synthetic R4 receptive fields with square grid spacing. (D) 26 synthetic R4 receptive fields random symmetrical spacing. Each panel is applied as a filter to 95 x 360 degree panoramas.
(PDF)

**S2 Fig. Natural scenes.** One example frame from each of the 33 panoramic videos of natural scenes. Videos were captured at various locations across the University of Sussex and adjacent

Stanmer Park. (top) 23 rotation only videos. For 3 locations examples were captured in open areas on both sunny and overcast days (total 6 panoramas). The remaining examples include trees either in a woodland or campus setting which occlude some or all of the sky. (bottom) 10 circling videos. A photograph of the Spidercam showing 4 cables connected to the camera assembly. Cable lengths are changed by winding or unwinding the cables from a spool using stepper motors, in order to move the camera in 3 dimensions. For all panoramas the performance group (high variance, low variance, or high failure), and the approximate distance to closest landmark is indicated on the y-axis.
(PDF)

**S3 Fig. Varying the Learning parameters.** (A) Logged mean error for the final rotation for all simulations across rotation only natural scenes, for combinations of parameters learning rate ($\eta$) and $\rho$ (see Methods for learning rule). Parameters $\eta = 0.01$ and $\rho = 0.06$ were selected from the dark blue region of low error (white star). Colourbar indicates logged mean error over all natural scenes in degrees. (B) Average error across all natural scenes when learning remains on or turned off during the probe trial (T-test p = 0.001). (C) Percentage of weights changed over multiple revolutions of learning for Angular Velocity = 60 deg/s, $\eta = 0.01$ and $\rho = 0.06$. (D) Developments in the weight space over 10 revolutions. After 5 revolutions the weight space is fully developed and very few changes occur.
(PDF)

**S4 Fig. Bump movement produced by different model parameters.** (A) In each row and column, the number of bump rotations for a single run of the model for a specific combination of weights is represented by colour (darker = more rotations; empty means weight combination was not viable). Bump rotations are measured by unwrapping the head direction estimate and finding the total degrees turned. On the x-axis, there are 7 sets of 7 columns. Each set of 7 columns has a fixed weight from R to EPG cell ranging from -0.5 to -5, shown on top x-axis. Within each set, the weight from $\Delta 7$ cells to EPG cells cycles through the same 7 values between -0.5 and -5, shown on the bottom x-axis. On the y axis, the weight from EPG to PEN cells varies every 10 sets of 10 rows (range 0.05 to 0.23 in steps of 0.02; shown on left y-axis) while the weight from PEN to EPG cells changes every cell (cycling in 10 steps from 0.05 to 0.23), shown on the right y-axis. Only parameters that meet the following conditions were considered viable: the bump must persist for the full simulation; the bump must be at least 2 cells wide and no wider than 8 cells; and the bump must move at least one full rotation within a 20000ms simulation. The parameter set in Table 1 results in 3 rotations in 20000ms. Selecting alternative parameters could allow different ranges of angular velocity to be accurately tracked. For all parameter combinations the weights from EPG to R, EPG to $\Delta 7$ and EPG to EPG are set to constant values specified in Table 1. (B) Number of rotations produced by the model when varying EPG to R weights (between 0.005 to 0.05 in steps of 0.005; x-axis). In each subpanel, the weight between two other groups of cells is varied (y-axis). The weights that were not varied were fixed to the values shown in Table 1.
(PDF)

## Author Contributions

**Conceptualization:** Rachael Stentiford, James C. Knight, Thomas Nowotny, Andrew Philippides, Paul Graham.

**Formal analysis:** Rachael Stentiford.

**Funding acquisition:** Thomas Nowotny, Andrew Philippides, Paul Graham.

**Investigation:** Rachael Stentiford.

**Methodology:** Rachael Stentiford, James C. Knight, Thomas Nowotny.

**Software:** Rachael Stentiford, James C. Knight.

**Supervision:** Thomas Nowotny, Andrew Philippides, Paul Graham.

**Writing – original draft:** Rachael Stentiford, Paul Graham.

**Writing – review & editing:** James C. Knight, Thomas Nowotny, Andrew Philippides, Paul Graham.

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
