## [Decision Letter · Decision Letter 0]

9 Apr 2024

Dear Dr Stentiford,

Thank you very much for submitting your manuscript "Estimating orientation in Natural scenes: A Spiking Neural Network Model of the Insect Central Complex" for consideration at PLOS Computational Biology.

As with all papers reviewed by the journal, your manuscript was reviewed by members of the editorial board and by several independent reviewers. In light of the reviews (below this email), we would like to invite the resubmission of a significantly-revised version that takes into account the reviewers' comments.

Both reviewers agree on the interest of the manuscript and the scientific topics it tackles. However, they raise significant issues that we would want to see addressed before any decision can be made about acceptation. First of all, we agree with the reviews that the manuscript would greatly benefit from a systematic exploration of the major parameters (weight pattern changes with increasing revolutions, rotation angular speed distribution, size of the stimulus triangles and distance between them, spacing pattern of the R2 & R4 neurons, input strength of the PEN neurons). Moreover, a couple of modelling choices or unclear results need to be more strongly supported, including the STDP rule used for inhibitory synapses or the error increase with PEN neuron input. Finally, we also agree with reviewer 2 that the manuscript would be significantly stronger with more model validation, for instance using visual stimuli that are frequently used in fly behavioral tests. 

We cannot make any decision about publication until we have seen the revised manuscript and your response to the reviewers' comments. Your revised manuscript is also likely to be sent to reviewers for further evaluation.

Sincerely,

Hugues Berry

Academic Editor

PLOS Computational Biology

Thomas Serre

Section Editor

PLOS Computational Biology

Reviewer's Responses to Questions

**Comments to the Authors:**

Reviewer #1: The paper presents a computational model focusing on orientation or heading direction within the insect central complex. While the core model shares similarities with several existing models in the literature, it incorporates more realistic visual inputs derived from recent experimental data (specifically, R2 and R4 neurons as used in this study). Tested with both simple stimuli (e.g. bright vertical bars or a linear array of triangles) and panoramic videos recorded from a moving camera in real environments, the model demonstrates its ability to estimate heading even with complex visual inputs, achieved through filtering by a small number of ring neurons with simple receptive fields, with the help of synaptic learning. While naturalistic visual inputs remain very limited, this represents a step forward. While the paper still exhibits some weaknesses, addressing them in a revision could further strengthen its contribution to the field.

Detailed comments:

The authors claim to introduce a novel spiking neural network model of the insect central complex. While the network itself may not be inherently novel given the anatomical constraints, the true innovation likely lies in a more realistic presentation of visual inputs. To better clarify the contribution of this paper, the authors should provide a more accurate description of the differences.

Regarding synaptic weight learning, the results presented are based on only a few revolutions. However, considering real animals may experience numerous rotations over a reasonable timeframe, it would be valuable to explore the mature weight patterns after learning from many revolutions.

The authors should consider systematically exploring the parameter space, including testing the effects of weight strength on the results.

Furthermore, changing the rotation angular speed or utilizing a naturalistic angular speed distribution may impact learning, particularly given the spiking-timing-dependent rule.

A closer examination of failed simulations for specific natural scenes where the activity bump was lost is warranted. Understanding the reasons behind these failures beyond correlational studies would provide valuable insights.

The effect of translational movement of the camera on the learning process appears to depend crucially on the distance from the camera to the landmarks. Therefore, documenting the distribution of distances from the camera to nearby objects is essential. Even crude estimates would prove useful for other researchers.

Fig. 2A:

some labels at the top (90 deg rotated) need more spacing

Need consistent notation: label D7 in Fig. 2 Vs delta 7 in Fig. 1

Fig 5:

B: Why are there pixel correlations at 180 deg? Bias in the video? It may cause confound with the ring activity correlation.

The organization of the panel labels is confusing. For example, “G: High correlations between ring neuron activations…” seems wrong.

Reviewer #2: The authors propose a ring-neuron network model with synthetic receptive fields with STDP (spike-timing dependent plasticity) for the insect central complex. They demonstrate that their model is able to reproduce observed neural activity and behavior under simple and natural scenes. Previously, neural network models of this brain area mostly focused on the function of the attractor dynamics and how a bump can be formed under simple stimuli. Other models showed how plastic ring circuits can form a map between visual stimuli and the ellipsoid body. This work fills the gap and provides a more complete picture of the mechanism of visual orientation of fruit flies. However, there are several critical issues the authors need to address. Here are my specific comments:

Major comments

1. The hypothesis of the hexagonal arrangement of the receptive fields of the R2 and R4 neurons seems crucial for the model. Is there any experimental evidence for the hypothesis? Otherwise, the authors should also test other arrangements, such as square lattice or random spacing. If the hexagonal arrangement delivers the best performance, this can be a testable prediction of the model.

2. The authors referred to Vogels et al 2011 for their model of plasticity for inhibitory synapses. However, unless I understand it incorrectly, the plasticity model in Vogels et al 2011 (Eqs 3-5 in Supporting Material) exhibits an opposite effect to what the authors modeled in the manuscript. Vogels et al 2011 used conductance-based synapses. Their weights represent synaptic conductance and are always positive. When postsynaptic and presynatpic spikes occur within a small time window, the weight increases, leading to strengthened inhibitory synapses. However, in this manuscript, the authors appeared to use current-based synapses, so the inhibitory weights are negative. When coincident spikes occur, the weights become less negative, or the inhibitory synapses are weakened. Can the authors elaborate? Is there any experimental support for this uncommon STDP rule?

3. It appears that the model has the best performance when the bump is solely driven by the visual input from the ring neurons and the error increases when the model takes additional input from PEN neurons (see Fig 3A, 4B, 5D & 5J). Why is that? Did the authors test whether they had a correct PEN input strength? Because the moving speed of the bump is also modulated by the PEN input, if the input strength is too high or too low, it might interfere with the ring-neuron-driven bump movement.

4. What happens if the virtual scene has a bright (white) background with one or two black stripes? This type of visual stimuli has been used widely in fruit fly behavioral tests. Flies are shown to navigate and remember the location of the stripes without problem. Does the model proposed by the authors perform equally well or is it unable to keep the bump due to the strong activation of most ring neurons? I would guess the latter, but I could be wrong. This can be a good way to validate the model.

5. It is a cool idea to test the model using triangles with different arrangements and the result is very impressive. However, not being able to discriminate between the objects (as tested in Ernst & Heisenberg 1999) is not equivalent to not being able to keep orientation (as shown in this study). It is likely that they are two completely different neural mechanisms responsible by different neural pathways. The authors need to acknowledge and discuss this in the Discussion section, or simply remove this part.

6. Following the comment above, I guess the inability to move the bump (Figure 4C) may be just a coincidence stemming from the specific sizes and arrangements of the cues and the hexagonal receptive fields. What happens if the size of the triangles and the space between them are changed? My main point is that if the authors want to keep this part, they should investigate the mechanism underlying such indiscriminability rather than just showing the phenomenon.

Minor comments

1. Line 113. Please specify the unit of tau. If the unit is millisecond, 50 ms for inhibitory synapses is way larger than typical GABAa receptors, unless the authors modeled GABAb. 100 ms for the excitatory synapses are also larger than what would be expected for cholinergic receptors. Is there any support for such large synaptic time constants? Is there any specific reason why excitatory synapses need longer time constants than that of inhibitory synapses? I can understand that longer time constants are crucial for stabilizing the dynamics of a neural circuit with a small number of neurons. The authors need to explain the reason in Methods or discuss it in Discussion.

2. Figure 2 legend, third line from bottom: missing the figure number in “(see Figure )”

3. Figure 2B. Please label the neuron indices (all or a few representative ones). Otherwise, without known which neuron is which, the connection matrices in panel C are meaningless.

4. Line 263: The phrase “median active EPG cell” is confusing. Using the word “median” makes people think that the author sorted the EPG cells by their activity level and choose the middle one.

5. Lines 323-333. Some sentences are too long and poorly written. Please rewrite them.

6. Line 332. “…resulting in multiple peaks.” Did the author mean several activity bumps in EPG cells? I do not see them in the figure.

7. Line 352. “Is” it.

8. Line 358. …in the error “measured” across all simulations….

9. Line 359. “… greater than 1% failed simulations.” Do the authors mean that they run more than 100 trials and count the number of failed trials? What do they define a "failed simulation?"

10. Figure 6. The authors may want to display a few example scenes for high, low variences and failed simulations.

11. The authors use the phrase “multi-polar ring neurons” several times in the text. However, it is confusing because “uni-polar”, “bi-polar” and “multi-polar” are usually used to describe the morphology of a neuron. The authors may want to find a better phrase for what they want to describe.

12. Line 378. “However”  should start a new sentence.

13. Line 378. “…all groups…” What groups? High, low variances and failed simulations?

14. Line 277-382. Again, a very long sentence hard to comprehend. Please rewrite it.

**Have the authors made all data and (if applicable) computational code underlying the findings in their manuscript fully available?**

Reviewer #1: Yes

Reviewer #2: **No: **Unless I miss it, I do not see the authors mentioning any link to their code in the manuscript.

PLOS authors have the option to publish the peer review history of their article (what does this mean?). If published, this will include your full peer review and any attached files.

Reviewer #1: No

Reviewer #2: **Yes: **Chung-Chuan Lo
---

## [Decision Letter · Decision Letter 1]

24 Jul 2024

Dear Dr Stentiford,

We are pleased to inform you that your manuscript 'Estimating orientation in Natural scenes: A Spiking Neural Network Model of the Insect Central Complex' has been provisionally accepted for publication in PLOS Computational Biology.

Best regards,

Hugues Berry

Section Editor

PLOS Computational Biology

Thomas Serre

Section Editor

PLOS Computational Biology

Reviewer's Responses to Questions

**Comments to the Authors:**

Reviewer #2: The authors have addressed all my comments and I would like to recommend the manuscript for publication.

**Have the authors made all data and (if applicable) computational code underlying the findings in their manuscript fully available?**

Reviewer #2: Yes

PLOS authors have the option to publish the peer review history of their article (what does this mean?). If published, this will include your full peer review and any attached files.

Reviewer #2: **Yes: **Chung-Chuan Lo

---

## [Editor Report · Acceptance letter]

10 Aug 2024

PCOMPBIOL-D-24-00265R1 

Estimating orientation in Natural scenes: A Spiking Neural Network Model of the Insect Central Complex

Dear Dr Stentiford,

I am pleased to inform you that your manuscript has been formally accepted for publication in PLOS Computational Biology. Your manuscript is now with our production department and you will be notified of the publication date in due course.

With kind regards,

Zsofia Freund
